# A Functional Extension of Semi-Structured Networks

**David Rügamer**
Department of Statistics, LMU Munich
Munich Center for Machine Learning (MCML)
Munich, Germany
david@stat.uni-muenchen.de

**Bernard X.W. Liew, Zainab Altai**
School of Sport, Rehabilitation and Exercise Sciences
University of Essex
Colchester, UK
[bl19622,z.altai]@essex.ac.uk

**Almond Stöcker**
Institute of Mathematics
École Polytechnic Fédéral de Lausanne (EPFL)
Lausanne, Switzerland
almond.stoecker@epfl.ch

## Abstract

Semi-structured networks (SSNs) merge the structures familiar from additive models with deep neural networks, allowing the modeling of interpretable partial feature effects while capturing higher-order non-linearities at the same time. A significant challenge in this integration is maintaining the interpretability of the additive model component. Inspired by large-scale biomechanics datasets, this paper explores extending SSNs to functional data. Existing methods in functional data analysis are promising but often not expressive enough to account for all interactions and non-linearities and do not scale well to large datasets. Although the SSN approach presents a compelling potential solution, its adaptation to functional data remains complex. In this work, we propose a functional SSN method that retains the advantageous properties of classical functional regression approaches while also improving scalability. Our numerical experiments demonstrate that this approach accurately recovers underlying signals, enhances predictive performance, and performs favorably compared to competing methods.

## 1  Introduction

Incorporating additive structures in neural networks to enhance interpretability has been a major theme of recent advances in deep learning. For example, neural additive models (NAMs; [1]) allow users to learn basis functions in an automatic, data-driven fashion using feature subnetworks and thereby provide an alternative modeling option to basis function approaches from statistics such as generalized additive models (GAMs; [17]). Various extensions and related proposals have been published in recent years. Notable examples include the joint basis function learning by [41] or the combination of GAMs and neural oblivious decision ensembles [8]. When combining structural assumptions with arbitrary deep architectures, the resulting network is often referred to as wide-and-deep [10] or semi-structured network (SSN; [49, 52]). Various extensions of SSNs have been proposed in recent years, including SSNs for survival analysis [24, 25], ordinal data fitting [22], for

38th Conference on Neural Information Processing Systems (NeurIPS 2024).

uncertainty quantification [2, 12], Bayesian SSNs [11], or SSNs incorporated into normalizing flows [23]. One of the key challenges in SSNs is to ensure the identifiability of the additive model part in the presence of a deep network predictor in order to maintain interpretability.

In this work, motivated by a large-scale biomechanical application, we study how to efficiently transfer these two concepts — the embedding of basis functions into neural networks and the extension to SSNs — for functional data analysis.

**Functional data analysis** Functional data analysis (FDA; [42]) is a research field of increasing significance in statistics and machine learning (e.g., [3, 37, 38, 57]) that extends existing modeling approaches to address the functional nature of some data types (e.g., sensor signals, waves, or growth curves). One well-known extension is the class of functional additive models [15, 53], which represent the functional analog of GAMs. These models adopt the classical regression model point of view but allow for both in- and outputs to be (discretized) functions. Functional additive models, however, are often not expressive enough to account for all interactions and non-linearities and do not scale well to large datasets.

**Biomechanics** A prominent example of a research field dealing with functional data is biomechanics. As in many other applications concerned with physical or biological signals, the integration of machine learning in biomechanics research offers many advantages (see, e.g., [26, 29, 30]). The research on joint moments in biomechanics, for example, provides insight into muscular activities, motor control strategies, chronic pain, and injury risks (see, e.g., [13, 31, 36, 47, 56]). By combining kinematic features with machine and deep learning to predict joint moments, researchers can bypass the need to gather high-quality biomechanics data and instead use "cheap" sensor data obtained through, e.g., everyday mobile devices which in turn allow predicting the "expensive" signals that can only be obtained in laboratory setups [20, 32, 33]. While a promising research direction, existing methods face various challenges including a lack of generalization and a missing clear understanding of the relationship learned.

## 1.1 Related work in functional data analysis

In FDA, there has been a successful translation of many different methods for scalar data to function-valued data, including functional regression models and functional machine learning techniques.

**Functional regression models** Regression models with scalar outputs and functional inputs are called scalar-on-function regression models. An overview of such methods can be found in [45]. Models with functional output and scalar inputs are called function-on-scalar models [9, 15]. They can be applied when the outcome or error function is assumed to be repeated realizations of a stochastic process. Combining these two approaches yields the function-on-function regression model with functions as input and output [see, e.g., 35, 53]. We provide a more technical description of function-on-function regression is given in Section 2.1.

**Functional machine learning** In recent years, several machine learning approaches for functional data have been proposed. One is Gaussian process functional regression [GPFR; 21, 54] which combines mean estimation and a multivariate Gaussian process for covariance estimation. As for classical Gaussian process regression, the GPFR suffers from scalability issues. Gradient boosting approaches for functional regression models have been proposed by [4, 6]. These approaches result in sparse and interpretable models, rendering them especially meaningful in high dimensions, but lack expressivity provided by neural network approaches. Pioneered by the functional multi-layer perceptron (MLP) of [48], various researchers have suggested different functional neural networks (FNNs) [16, 44, 55], including MLPs with functional outcome (cf. Section 2.2 for technical details). A recent overview is given in [59]. Existing approaches are similar in that they learn a type of embedding to represent functions in a space where classical computations can be applied (cf. Fig. 2(a)).

## 1.2 Current limitation and our contribution

From the related literature, we can identify the most relevant methods suitable for our modeling challenge. Neural network approaches such as the functional MLP provide the most flexible class for functional data, whereas additive models and additive boosting approaches such as [6, 53] yield interpretable models. An additive combination of these efforts to obtain both an interpretable model part and the possibility of making the model more flexible is hence an attractive option.

**Open challenges** While semi-structured models are suited for a combination of interpretable additive structures and deep neural networks, existing approaches cannot simply be transferred for application with functional data. Due to the structure and implementation of SSNs, a naïve implementation would neither be scalable to large-scale datasets nor is it clear how to incorporate identifiability constraints to ensure interpretability of the structured functional regression part in the presence of a deep neural network.

**Our contributions** In this work, we address these limitations and propose an SSN approach for functional data. Our method is not only the first semi-structured modeling approach in the domain of functional data, but also presents a more scalable and flexible alternative to existing functional regression approaches. In order to preserve the original properties of functional regression models, we further suggest an orthogonalization routine to solve the accompanied identifiability issue.

## 2 Notation and background

We assume functional inputs are given by $J \geq 1$ second-order, zero-mean stochastic processes $X_j$ with square integrable realizations $x_j : \mathcal{S}_j \to \mathbb{R}$, i.e. with $x_j \in L^2(\mathcal{S}_j), j = 1, \ldots, J$ on real intervals $\mathcal{S}_j$. Similarly, a functional outcome is given by a suitable stochastic process $Y$ over a compact set $\mathcal{T} \subset \mathbb{R}$ with realizations $y : \mathcal{T} \to \mathbb{R}$, $y \in L^2(\mathcal{T}) =: \mathcal{G}$. For the functional inputs, we simplify notation by combining all (random) input functions into a vector $X(\boldsymbol{s}) = (X_1(s_1), \ldots, X_J(s_J))$ with realization $x(\boldsymbol{s}) \in \mathbb{R}^J$, where $\boldsymbol{s} := (s_1, \ldots, s_J) \in \mathcal{S} \subset \mathbb{R}^J$ is a $J$-tuple from domain $\mathcal{S} := \mathcal{S}_1 \times \cdots \times \mathcal{S}_J$, and $\mathcal{H} := L^2(\mathcal{S}_1) \times \cdots \times L^2(\mathcal{S}_J)$.

Next, we introduce function-on-function regression (FFR) and establish a link to FNNs to motivate our own approach.

### 2.1 Function-on-function regression

An FFR for the expected outcome $\mu(t) := \mathbb{E}(Y(t)|X)$ of $Y(t), t \in \mathcal{T}$ using inputs $X$ can be defined as follows:

$$\mathbb{E}(Y(t)|X = x) = \tau \left( b(t) + \sum_{j=1}^{J} \int_{\mathcal{S}_j} w_j(s,t) x_j(s) ds \right), \tag{1}$$

where $b(t)$ is a functional intercept/bias and the $w_j(s,t)$ are weight surfaces describing the influence of the $j$th functional predictor at time point $s \in \mathcal{S}_j$ on the functional outcome at time point $t \in \mathcal{T}$. $\tau$ is a point-wise transformation function, mapping the affine transformation of functional predictors to a suitable domain (e.g., $\tau(\cdot) = \exp(\cdot)$ to obtain positive values in case $Y$ is a count process). Various extensions of the FFR model in (1) exist. Examples include time-varying integration limits or changing the linear influence of $x(t)$ to a non-linear mapping.

**Example** Figure 1 shows an example of a function-on-function regression by visualizing the corresponding learned weight surface, with three highlighted areas: a) An isolated positive weight multiplied with a positive feature signal at $x(90)$ induces a small spike at $y(10)$. b) A more extensive negative weight multiplied with mostly positive feature values for $s \in [1, 20]$ and integrated over all time points results in a bell-shaped negative outcome signal at around $t = 30$. c) A large but faded positive weight region multiplied with positive feature values results in a slight increase in the response function in the range $t \in [50, 100]$. In most applications of FFR, the goal is to find these salient areas in the weight surface to better understand the relationship between input and output signal.

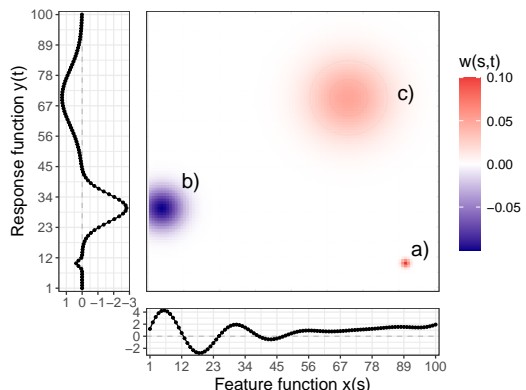

Figure 1: Exemplary weight surface (center), feature signal (bottom), and the resulting response signal (left) when integrating $\int x(s)w(s,t)ds$.

## 2.2 Functional neural networks

To establish the connection between FFR and FNNs, it is instructive to consider a function-on-function MLP (FFMLP). In its basic form, an MLP for scalar values consists of neurons $h : \mathbb{R} \to \mathbb{R}, x \mapsto \tau(b + w^\top x)$ arranged in multiple layers, each layer stacked one on top of the other. The extension to a fully functional $L$-layer MLP can be defined recursively by the $k$th output neuron $h_k^{(l)}$ of a functional layer $l = 1, \ldots, L$ as

$$h_k^{(l)}(t) = \tau^{(l)} \left( b_k^{(l)}(t) + \sum_{m=1}^{M_{l-1}} \int w_{m,k}^{(l)}(s,t) h_m^{(l-1)}(s) ds \right), \qquad (2)$$

where $M_l$ denotes the number of neurons in layer $l$, $b_k^{(l)} \in L^2(\mathcal{U}_m^{(l)})$ and $w_{m,k}^{(l)} \in L^2(\mathcal{U}_m^{(l-1)} \times \mathcal{U}_m^{(l)})$ for some functional domains $\mathcal{U}_m^{(l)}$, input layer $h_m^{(0)}(\cdot) = x_m(\cdot)$ for all predictors $m = 1, \ldots, J$, and last layer with $M_L = 1$ neuron $h^{(L)}(t) = \mathbb{E}(Y(t)|X)$.

# 3 Semi-structured functional networks

Our approach generalizes the previous models by considering a general neural network $\Lambda : \mathcal{H} \to \mathcal{G}$ that models the input-output mapping from the set of functional features $X$ to the expected outcome $\mu(t)$ of the functional response for $t \in \mathcal{T}$ as

$$\mu(t) = \Lambda(X)(t) = \tau \left( \lambda^+(X)(t) + \lambda^-(X)(t) \right), \qquad (3)$$

comprising an FFR part $\lambda^+(X)(t)$ and a deeper FNN architecture $\lambda^-(X)(t)$, which are added and transformed using an activation function $\tau$. This combination can also be thought of as a functional version of a residual connection. In the following, we drop the functional input arguments of $\lambda^+$ and $\lambda^-$ for better readability. The model in (3) combines structured interpretable models as described in Section 2.1 with an arbitrary deep network such as the one in Section 2.2. In particular, this allows for improving the performance of a simple FFR while retaining as much interpretability as possible (cf. Section 3.3).

**Interpretable model part** As in (1), we model the interpretable part $\lambda^+(t) = \sum_{j=0}^{J} \lambda_j^+(t)$ as a sum of linear functional terms $\lambda_j^+(t) = \int_{\mathcal{S}_j} w_j(s,t) x_j(s) ds$. To make our model more concrete, we can expand each weight surface $w_j$ in a finite-dimensional tensor-product basis as

$$w_j(s,t) = \boldsymbol{\psi}(t)^\top \boldsymbol{\Theta}_j \boldsymbol{\phi}_j(s) \qquad (4)$$

over fixed function bases $\boldsymbol{\phi}_j = \{\phi_{jk}\}_{k=1}^{K_j}$, $\phi_{jk} \in L^2(\mathcal{S}_j)$ and $\boldsymbol{\psi}(t) = \{\psi_u\}_{u=1}^{U}$, $\psi_u \in L^2(\mathcal{T})$ with weight matrix $\boldsymbol{\Theta}_j \in \mathbb{R}^U \times \mathbb{R}^{K_j}$. Analogously, we can represent the intercept as $\lambda_0^+(t) = b(t) = \boldsymbol{\psi}(t)^\top \boldsymbol{\Theta}_0$. We may understand the model's interpretable part for the $j$th feature as a functional encoder $\boldsymbol{\phi}_j^*(X_j) = \{\phi_{jk}^*(X_j) = \int \phi_{jk}(s) X_j(s) ds\}_{k=1}^{K_j}$, encoding $X_j$ into a latent variable $\boldsymbol{z}_j \in \mathbb{R}^{K_j}$, and a linear decoder $\boldsymbol{\psi}$ with weights $\boldsymbol{\Theta}_j$ mapping $\boldsymbol{z}_j$ to the function $\lambda_j^+(t)$:

$$X_j \xmapsto{\boldsymbol{\phi}_j^*} \boldsymbol{z}_j \xmapsto{\boldsymbol{\Theta}_j \boldsymbol{\psi}} \lambda_j^+.$$

Here, $\boldsymbol{\phi}_j^*$ presents the dual basis to $\boldsymbol{\phi}_j$. Visually, the weight surface $w_j$ in (4) can be interpreted as exemplarily shown in Figure 1.

**Deep model part** The part $\lambda^-(t)$ in (3) is a placeholder for a more complex network. For FNNs, a conventional neural network might be applied after encoding $X$ with $\phi^*$ and before decoding its results with $\psi$, such that en- and decoding are shared with the interpretable part $\lambda^+$ as depicted in Fig. 2a. Alternatively, a general FFMLP as described in Section 2.2 can be embedded into the approach in Fig. 2a. As researchers have often already found well-working architectures for their data, a practically more realistic architecture for semi-structured FNNs is to allow for a more generic deep model as depicted in Fig. 2b. This is also the case in our application on biomechanical sensor data in Section 4.2. Here, we choose $\lambda^-(t)$ to be a specific InceptionTime network architecture [19] that is well-established in the field and train its weights along with the weights $\boldsymbol{\Theta}$ of $\lambda^+(t)$.

## 3.1 Implementation for discretized features

As functional predictors are usually only observed on a discretized grid on the domains $\mathcal{S}_j$, we now derive a way to implement $\lambda^+(t)$ in practice. Depending on the architecture used for the deep part, we might proceed analogously with $\lambda^-(t)$. Let $x_j^{(i)}(s_r) \in \mathbb{R}$ be the evaluation of the $i$th realization $x_j^{(i)}$ of $X_j$ at time points $s_r, r = 1, \ldots, R$, stacked into a vector $\boldsymbol{x}_j \in \mathbb{R}^R$. For better readability, we assume that these time points are equal across all $J$ features. With only discrete evaluations available, we approximate integrals over $\mathcal{S}_j$ numerically with integration weights $\Delta(s_r)$, e.g., using trapezoidal Riemann weights. Given the tensor-product basis representation of the weight surface $w_j(s, t)$ in (4), we effectively evaluate $\boldsymbol{\phi}_j(s_r)$ for all $R$ time points, yielding $\boldsymbol{\Phi}_j = [\boldsymbol{\phi}_j(s_1), \ldots, \boldsymbol{\phi}_j(s_R)] \in \mathbb{R}^{K_j \times R}$, and obtain the row-vector $\boldsymbol{\Phi}_j^*$ of approximate functionals

$$\phi_{jk}^*(x_j) = \int \phi_{jk}(s)x_j(s)ds \approx \sum_{r=1}^{R} \Delta_j(s_r)\phi_{jk}(s_r)x_j(s_r)$$

as

$$\boldsymbol{\Phi}_j^* = (\boldsymbol{\Delta}_j \circ \boldsymbol{x}_j)\boldsymbol{\Phi}_j^\top \in \mathbb{R}^{1 \times K_j},$$

where $\circ$ denotes the Hadamard product and $\boldsymbol{\Delta}_j = [\Delta_j(s_1), \ldots, \Delta_j(s_R)]^\top$. Putting everything together, we can represent the $j$th interpretable functional model term $\lambda_j^+(t)$ as

$$\lambda_j^+(t) = \int_{\mathcal{S}_j} x_j(s)w_j(s, t)ds \approx \boldsymbol{\Phi}_j^* \boldsymbol{\Theta}_j \boldsymbol{\psi}(t). \tag{5}$$

This model part is still a function over the outcome domain $\mathcal{T}$, but discretized over the predictor domains. We can now proceed with the optimization when the functional outcome is represented with finitely many observations.

## 3.2 Optimization for discretized outcomes

For the $i$th observation, we can measure the goodness-of-fit of our model $\mu(t)$ by taking the point-wise loss function $l(y^{(i)}(t), \hat{\mu}^{(i)}(t))$ of the $i$th realized function $y^{(i)}$ and the predicted outcome $\hat{\mu}^{(i)}$ given $\boldsymbol{x}^{(i)}$ and integrate over the functional domain to obtain the $i$th functional loss contribution $\ell$, i.e.,

$$\ell(y^{(i)}, \hat{\mu}^{(i)}) = \int_{\mathcal{T}} l(y^{(i)}(t), \hat{\mu}^{(i)}(t))dt. \tag{6}$$

The overall objective, our empirical risk, is then given by the sum of (6) over all $n$ observations. In practice, integrals in (6) are not available in closed form in general and the functional outcome is only observed on a finite grid. Therefore, similar to the discretization of the weight surface, let $y^{(i)}(t_q)$ be the observations of the outcome for time points $t_q \in \mathcal{T}, q = 1, \ldots, Q$, summarized for all time points as $\boldsymbol{y}^{(i)} = (y^{(i)}(t_1), \ldots, y^{(i)}(t_Q))$. Given a dataset $\mathcal{D}$ with $n$ observations of these functions, i.e., $\mathcal{D} = \{(\boldsymbol{x}^{(i)}, \boldsymbol{y}^{(i)})\}_{i=1,\ldots,n}$, our overall objective then becomes

$$\min \sum_{i=1}^{n} \sum_{q=1}^{Q} \Xi(t_q)l(y^{(i)}(t_q), \hat{\mu}^{(i)}(t_q)), \tag{7}$$

where $\Xi(\cdot)$ are integration weights.

As it less efficient to represent $\mu$ as an actual function of $t$ if $Y$ is only given for fixed observed time points $t_q$, we can adapt (5) to predict a $Q$-dimensional vector using

$$\boldsymbol{\lambda}_j^+ \approx \boldsymbol{\Phi}_j^* \boldsymbol{\Theta}_j \boldsymbol{\Psi} \in \mathbb{R}^{1 \times Q} \tag{8}$$

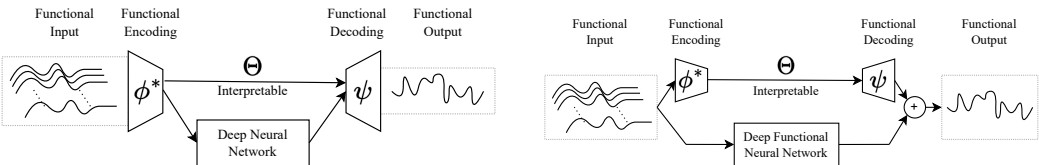

(a) A semi-str. FNN with shared en- and decoding.  (b) A semi-str. FNN with shared in- and output layer.

Figure 2: Different semi-structured architectures.

where $\boldsymbol{\Psi} = [\boldsymbol{\psi}(t_1), \dots, \boldsymbol{\psi}(t_Q)]$. The deep network $\lambda^-$ naturally yields the same-dimensional output when sharing the decoder. If a flexible deep model is used for $\lambda^-$ as in Fig. 2b, $\boldsymbol{\lambda}^-$ can be defined with $Q$ output units to match the dimension of $\lambda^+$. For one observation, the objective (7) can now be written as a vector-based loss function of $\boldsymbol{y}^{(i)} \in \mathbb{R}^{1 \times Q}$ and $\boldsymbol{\mu}^{(i)} = \sum_j \boldsymbol{\lambda}_j^{+(i)} + \boldsymbol{\lambda}^{-(i)} \in \mathbb{R}^{1 \times Q}$, or, when stacking these vectors, as loss function between two $n \times Q$ matrices.

## 3.3 Post-hoc orthogonalization

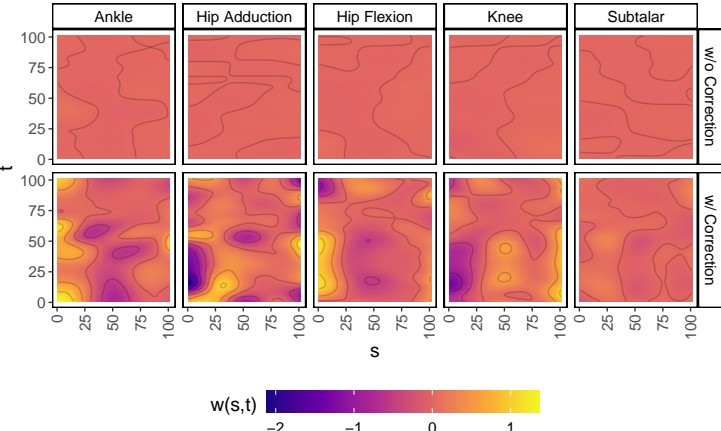

Figure 3: Estimated weight surfaces of the one functional shank gyroscope predictor in $\lambda^+$ for the different joints (columns), before and after correcting with the orthogonalization (rows). The color of each surface corresponds to the weight a predictor sensor signal at time points $s$ (x-axis) is estimated to have on the $t$th time point (y-axis) of the outcome (cf. Fig. 1). Without correction (upper row), the interpretable model part is not only incorrectly estimated but no effect at all is attributed to it. After correction, distinct patterns for some of the time point combinations and joints are visible, e.g., suggesting that an increased gyroscope value for early time points $s$ has a negative influence (dark purple color) on the first half of the hip adduction moment (bottom row, second plot from the left).

Since $\lambda^-$ is a potentially very expressive deep neural network that can also capture all functional linear dependencies modeled by $\lambda^+$, the interpretable model part of the semi-structured FNN is not identifiable without further constraints. Non-identifiability, in turn, means that we cannot simply interpret the weights of the interpretable model part as it is not clear how much these are diluted by the deep network part. This can, e.g., be seen in Figure 3 which shows selected weight surfaces from our biomechanical application in Section 4. In this case, the deep network (an InceptionNet) captures almost all linearity of the structured part (upper row) and only after correction do we obtain the actual weight surfaces. We achieve this by extending the post-hoc orthogonalization [PHO; 49] for functional in- and outputs. First, we rewrite $\lambda^+(t) \in \mathbb{R}$, the sum of all $J$ terms in (5), as

$$\lambda^+(t) = \text{vec}(\lambda^+(t)) \overset{(5)}{=} \text{vec}\left(\sum_{j=1}^{J} \boldsymbol{\Phi}_j^* \boldsymbol{\Theta}_j \boldsymbol{\psi}(t)\right) = \sum_{j=1}^{J} (\boldsymbol{\psi}(t)^\top \otimes \boldsymbol{\Phi}_j^*) \text{vec}(\boldsymbol{\Theta}_j) =: \sum_{j=1}^{J} \boldsymbol{\Omega}_j(t) \boldsymbol{\theta}_j$$

with $\boldsymbol{\Omega}_j(t) = (\boldsymbol{\psi}(t)^\top \otimes \boldsymbol{\Phi}_j^*) \in \mathbb{R}^{1 \times K_j \cdot U}$, where $\otimes$ denotes the Kronecker product, and $\boldsymbol{\theta}_j = \text{vec}(\boldsymbol{\Theta}_j) \in \mathbb{R}^{K_j \cdot U}$. This reformulation allows us to represent the functional interpretable model part as a linear combination $\boldsymbol{\Omega}(t)\boldsymbol{\theta} = [\boldsymbol{\Omega}_1(t), \dots, \boldsymbol{\Omega}_J(t)][\boldsymbol{\theta}_1^\top, \dots, \boldsymbol{\theta}_J^\top]^\top$ of basis products and their coefficients, and thereby to apply PHO for semi-structured FNNs. Let $\boldsymbol{\Omega}_{\{N\}}$ and $\boldsymbol{\lambda}_{\{N\}}$ be the stacked matrices for the $N = n \cdot Q$ data points in the training set $\mathcal{D}$ after stacking the $\boldsymbol{\Omega}(t)$ and $\boldsymbol{\lambda}^-(t)$ for all $Q$ time points of all $n$ observed curves. Then we can obtain an identifiable and thus interpretable functional model part from the trained semi-structured FNN with basis matrix $\boldsymbol{\Omega}_{\{N\}}$, parameters $\boldsymbol{\theta}$, and deep network part $\boldsymbol{\lambda}_{\{N\}}^-$ by computing the following:

$$\widetilde{\boldsymbol{\theta}} = \boldsymbol{\theta} + \boldsymbol{\Omega}_{\{N\}}^\dagger \boldsymbol{\lambda}_{\{N\}}^- \quad \text{and} \quad \boldsymbol{\lambda}_{\{N\}}^\perp = \mathcal{P}_{\boldsymbol{\Omega}_{\{N\}}}^\perp \boldsymbol{\lambda}_{\{N\}}^-, \tag{9}$$

where $\widetilde{\boldsymbol{\theta}}$ is the corrected basis coefficient vector, $\boldsymbol{\Omega}_{\{N\}}^\dagger$ is the Moore-Penrose pseudoinverse of $\boldsymbol{\Omega}_{\{N\}}$, $\mathcal{P}_{\boldsymbol{\Omega}_{\{N\}}}^\perp$ is a projection matrix projecting into the orthogonal complement spanned by columns of

$\mathbf{\Omega}_{\{N\}}$, and $\boldsymbol{\lambda}_{\{N\}}^{\perp}$ are the corrected deep network predictions orthogonal to the interpretable part. Using $\widetilde{\boldsymbol{\theta}}_j$ from $\widetilde{\boldsymbol{\theta}} = [\widetilde{\boldsymbol{\theta}}_1, \ldots, \widetilde{\boldsymbol{\theta}}_J]$ instead of the original $\boldsymbol{\theta}_j$s will yield updated weights $\widetilde{w}_j$ for the surfaces from (4) of the interpretable functional model part, which can be interpreted irrespective of the presence of a deep network.

### 3.4 Scalable implementation

When implementing semi-structured FNNs, $\lambda^-$ can be chosen arbitrarily to the needs of the data modeler. Hence, we do not explicitly elaborate on a particular option here and assume an efficient implementation of this part. Instead, we focus on a careful implementation of the structured model part to avoid unfavorable scaling as in other existing implementations (cf. Figure 4(b)). To mitigate this problem (in particular reducing the space complexity), we propose two implementation tricks. While these mainly reduce the complexity scaling w.r.t. $J$ and $R$, implementing an FFR in a neural network already yields an improvement by allowing to cap the memory costs to a fixed amount through the use of mini-batch training. In contrast, a full-batch optimization routine of (7) as implemented in classic approaches (e.g., [5, 53]) scales both with $n$ and $Q$ as data is transformed in a long-format, which can grow particularly fast for sensor data. For example, only $n = 1000$ observations of a functional response evaluated at $Q = 1000$ time points already results in a data matrix of 1 million rows. Classic implementations also scale unfavorably in terms of the number of features $J$ and observed times points $R$ for these functions. In the case where all functional predictors are measured on the same scale, fitting an FFR model requires inverting a matrix of size $(n \cdot Q) \times ((\sum_{j=1}^{J} K_j) \cdot U)$, which becomes infeasible for larger amounts of predictors. Even setting up and storing this matrix can be a bottleneck.

**Array computation** In contrast, when using the representation in (5), the basis matrices in $s$- and $t$-dimension (i.e. for in- and output) are never explicitly combined into a larger matrix, but set up individually, and a network forward-pass only requires their multiplication with the weight matrix $\boldsymbol{\Theta}$. As the matrix $\boldsymbol{\Psi}$ is the same for all $J$ model terms, we can further recycle it for these computations.

**Basis recycling** In addition, if some or all predictors in $X$ share the same time domain and evaluation points $s_1, \ldots, s_R$, we can save additional memory by not having to set up $J$ individual bases $\boldsymbol{\Phi}_j$.

## 4 Numerical experiments

In the following, we show that our model is able to recover the true signal in simulation experiments and compares favorably to state-of-the-art methods both in simulated and real-world applications. As comparison methods, we use an additive model formulation of the FFR model as proposed in [53], an FFR model based on boosting [5] and a deep neural network without interpretable model part. For the latter and the deep part of the semi-structured model, we use a tuned InceptionTime [19] architecture from the biomechanics literature [32]. As FFR and boosting provide an automatic mechanism for smoothing, no additional tuning is required. In all our experiments, we use thin plate regression splines [60] but also obtained similar results with B-splines of order three and first-order difference penalties.

**Hypotheses** In our numerical experiments, we specifically investigate the following hypotheses:

- **Comparable performance with better scalability**: For additive models without the deep part, our model can recover complex simulated function-on-function relationships (**H1a**) with similar estimation and prediction performance as other interpretable models (**H1b**) while scaling better than existing approaches (**H1c**).

- **Favorable real-world properties**: In real-world applications, our model is on par or better in prediction performance with current approaches (**H2a**), with a similar or better resemblance when generating output functions (**H2b**), and provides a much more meaningful interpretation than the deep MLP (**H2c**). Further, we conjecture that our approach is better in prediction performance than its two (structured, deep) individual components alone (**H2d**).

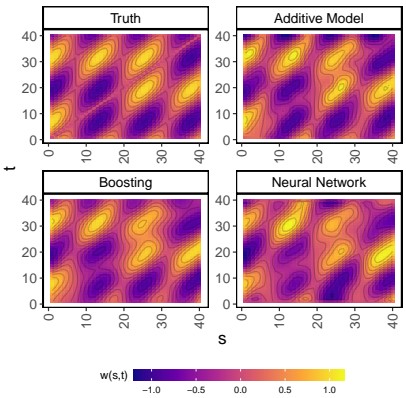

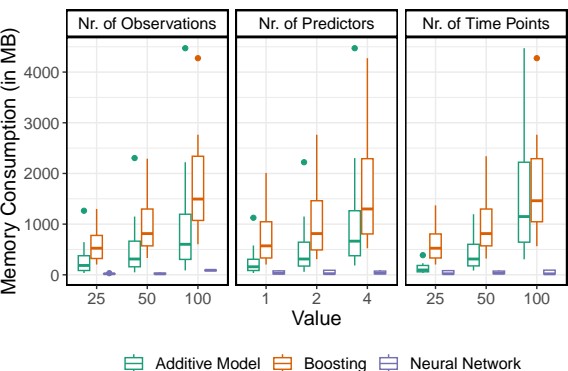

(a) True weight surface $w(s,t)$ used in the simulation study for large SNR along with estimation results of different methods.

(b) Memory consumption of different methods (colors) for different amounts of functional observations $n$ (left), functional predictors $J$ (center), and time points $R$ (right).

Figure 4: Simulation study results.

## 4.1 Simulation study

**Prediction and estimation performance (H1a,b)** For the model's estimation and prediction performance, we simulate a function-on-function relationship based on a complex weight relationship $w(s,t)$ between one functional predictor and a functional outcome. Figure 4(a) depicts this true weight along with an exemplary estimation by the different methods. The estimation performance of the weight surface and all methods' prediction performance over 30 different simulation runs each with a signal-to-noise ratio SNR $\in \{0.1, 1\}$ and $n \in \{320, 640, 1280\}$ functional observations is summarized in Figure 6 in the Appendix. While the models perform on par in most cases, we find that the network works better for more noisy settings (SNR = 0.1) and is not as good as other methods for higher signal-to-noise ratio (as also shown in Figure 4(a)), but with negligible differences. Overall, there are only minor differences in both estimation and prediction performance. This result suggests that our method works just as well as the other methods for estimating FFR.

**Scalability (H1c)** To investigate the scalability of all methods, we subsample the data from the Carmago study presented in the following subsection by reducing observations $n$, features $J$, and/or the number of observed time points $R$. More specifically, we investigate the memory consumption of the three implementations previously discussed (additive model, boosting, neural network) for $n \in \{25, 50, 100\}$ functional observations, $Q = R \in \{25, 50, 100\}$ observed data points, and $J \in \{1, 2, 4\}$ functional features. We restrict this study to rather small functional datasets as the memory consumption of other methods is superlinear. This can also be seen in Figure 4(b) presented earlier, which summarizes the result of this simulation study and clearly shows the advantage of our implementation compared to existing ones form [5, 53].

## 4.2 Real-world datasets

Our method being originally motivated by applications in the field of biomechanics, we now analyze two real-world datasets with biomechanical sensor data. In addition to these applications in biomechanics, we provide further applications to EMG-EEG synchronization, air quality prediction and hot water consumption profiles in Appendix C to demonstrate the versatility of our approach.

### 4.2.1 Fukuchi and Liew datasets (H2a)

The data analyzed in the first experiment is a collection of three publicly available running datasets [14, 27, 28]. A recent study of [32] used this data set to show the potential of deep learning approaches for predicting joint moments during gait (in total 12 different outcome features). The given data set collection is challenging as it provides only $n = 490$ functional samples but $J = 27$ partially highly correlated functional features. We follow the preprocessing steps of [32] and split the data into 80% training and 20% test data for measuring performance according to the authors' split indices.

As predictors we use the gait cycle of all available joint measurements, that is the 3D joint angle, velocity, and acceleration of the bilateral ankle, knee, and hip joints. Predictors in both training and test datasets are separately demeaned and scaled using only the information from the training set. As in the simulation studies, we run the additive model, Boosting, and our approach and compare the performance across all 6 different outcome variables. We use the relative and absolute root mean squared error (RMSE) as suggested in [46] and the Pearson correlation coefficient [20] to measure performance. In contrast to our simulation study, the additive model implementation of FFR cannot deal with this large amount of functional features. We, therefore, split our analysis into one comparison of the additive model, an FNN, and a semi-structured FNN (SSFNN) on a selected set of features, and a comparison of SSFNN against boosting (which has an inbuilt feature selection).

Results in Table 1 summarize test performances across all 12 different outcome types. We see that the (SS)FNN approaches perform equally well or better compared to the additive model on the selected set of features, and SSFNN also matches the performance of Boosting when using all features.

Table 1: Median (and mean absolute deviation in brackets) of the relative RMSE across all outcome responses in the Fukuchi and Liew datasets for different methods (columns) using either selected (sel.) or all features (all). The best method is highlighted in bold.

|  | Add. Model (sel.) | FNN (sel.) | SSFNN (sel.) | Boosting (all) | SSFNN (all) |
|---|---|---|---|---|---|
| rel. RMSE ($\downarrow$) | 0.36 (0.08) | 0.31 (0.08) | **0.30** (0.03) | 0.30 (0.10) | **0.25** (0.04) |

#### 4.2.2 Carmago data set (H2a-c)

The second data set is a publicly available dataset on lower limb biomechanics in walking across different surfaces level-ground walking, treadmill walking, stair ascent, stair descent, ramp ascent, and ramp descent [7]. Due to the size of the data set consisting of 21787 functional observations, $Q = R = 101$ observed time points, and $J = 24$ predictors, it is not feasible to apply either additive model-based FFR or boosting. We, therefore, compare SSFNN to a (structured) neural-based FFR model and a deep-only neural network. Based on the pre-defined 70/30% split for training and testing, we run the models for each of the 5 outcomes (joints) and compute the relative MSE difference, a functional $R^2$ analogon with values in $(-\infty, 1]$ (see Ap-

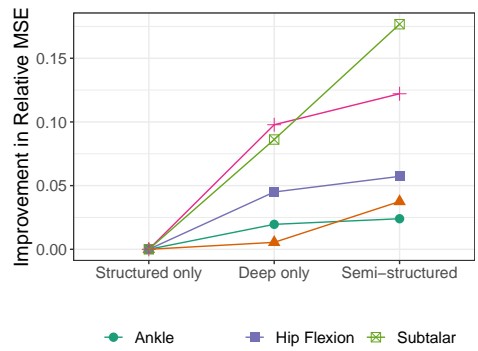

Figure 5: Comparison of performance improvements (larger is better) in mean squared error (MSE) for different joints (outcomes).

pendix A.1 for definition). Table 2 shows the results averaged for all outcome types while Figure 5 provides performance changes individually for the 5 different joints.

We observe that the semi-structured model performs better than the deep model, which in turn is better than the structured model. Figure 8 in the Appendix further depicts the predicted functions for all 5 outcomes using the 3 models. While for some joints the prediction of the semi-structured and deep-only model is very similar, the performance for the subtalar joint is notably better when combining a structured model and the deep neural network to form a semi-structured model.

Table 2: Mean performance (standard deviation in brackets) of the relative MSE difference (a value of 1 corresponds to the optimal model with zero error) across all five outcomes.

| Structured | Deep | Semi-Structured |
|---|---|---|
| 0.872 (0.094) | 0.923 (0.065) | **0.955 (0.030)** |

## 5   Discussion

Our proposed method is an extension of semi-structured networks to cater to functional data and deals with the issue of identifiability by proposing a functional extension of post-hoc orthogonalization. Using array computations and basis recycling the method also solves scalability issues present in existing methods. Our experimental results reveal that functional semi-structured networks yield similar or superior performance compared to other methods across varied biomechanical sensor datasets.

**Novelty** The approach proposed in Section 3 shows similarities to an autoeconder architecture. While a functional autoencoder has been proposed in the past [18] and most recently by [61], these studies focus on the representation learning (and reconstruction) of the same signal. Other papers focusing on function-on-function regression (e.g., [34, 43, 58]) suggest similar approaches to ours but without the option to jointly train a structured model and a deep network.

**Limitations and Future Research** While our model shows good performance on various datasets and in simulations, we believe that our approach can be further improved for applications on high-dimensional sensor data by incorporating appropriate sparsity penalties. Due to the functional nature of the data, feature selection would result in a great reduction of complexity and thereby potentially yield better generalization. Our general model formulation would further allow the extension to a non-linear FFR model $\lambda^-(t) = \sum_{j=1}^{J} \int_{\mathcal{S}_j} f_j(x_j(s), t) ds$ using smooth functions $f_j$ in three dimensions (feature-, $s$- and $t$-direction). Although this extension is still an additive model in the functional features, it is challenging to interpret the resulting additive effects.

## Acknowledgments and Disclosure of Funding

We thank the four anonymous reviewers and the area chair for providing valuable feedback.

DR's research is funded by the Deutsche Forschungsgemeinschaft (DFG, German Research Foundation) – 548823575. BL is funded by the Medical Research Council, UK (MR/Y013557/1) and Innovate UK (10093679). AS acknowledges support from SNSF Grant 200020_207367.

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

# A    Further details

## A.1    Functional $R^2$ definition

Let $n$ be the number of observed curves indexed with $i$. Given the $i$th observation of a function output signal $y^{(i)}(t)$ for $t \in \mathcal{T}$ with functional domain $\mathcal{T}$ and corresponding predicted value $\mu^{(i)}(t)$, we define the functional $R^2$ as

$$\varkappa(\{y^{(i)}, \mu^{(i)}\}_{i=1,\ldots,n}) := n^{-1} \sum_{i=1}^{n} \frac{\int_{\mathcal{T}}(y^{(i)}(t))^2 dt - \int_{\mathcal{T}}(y^{(i)}(t) - \mu^{(i)}(t))^2 dt}{\int_{\mathcal{T}}(y^{(i)}(t))^2 dt}. \tag{10}$$

Since $\int_{\mathcal{T}}(y^{(i)}(t) - \mu^{(i)}(t))^2 dt \geq 0$, the nominator is the smallest for $y(t) \equiv \mu(t)$, in which case $\varkappa = 1$. In contrast, if the prediction $\mu$ is $\mu \equiv 0$, $\varkappa = 0$. For models that perform worse than the constant zero predictor $\int_{\mathcal{T}}(y^{(i)}(t) - \mu^{(i)}(t))^2 dt$ can be larger than $\int_{\mathcal{T}}(y^{(i)}(t))^2 dt$, yielding a negative nominator and hence negative $\varkappa$ values.

## A.2    Penalization scheme

In order to enforce smoothness of estimated functional relationships in $\lambda^+$ in both $s$- and $t$-direction, we can employ a quadratic smoothing penalty. For regression splines with evaluated bases $\boldsymbol{\Phi}_j$ and $\boldsymbol{\Psi}$, smoothness is usually defined by some form of difference penalties, which we define to be encoded in the matrices $\boldsymbol{P}_s$ and $\boldsymbol{P}_t$, respectively. For the array computation, we want to penalize all differences in $s$-direction, which can be done by using

$$\text{pen}_s = \text{vec}(\boldsymbol{\Theta})^\top (\boldsymbol{I}_U \otimes \boldsymbol{P}_s)\text{vec}(\boldsymbol{\Theta}) = \text{vec}(\boldsymbol{\Theta} \circ (\boldsymbol{P}_s\boldsymbol{\Theta}))^\top \mathbf{1}.$$

Analogously, the $t$-direction can be penalized as

$$\text{pen}_t = \text{vec}(\boldsymbol{\Theta})^\top (\boldsymbol{P}_t \otimes \boldsymbol{K}_j)\text{vec}(\boldsymbol{\Theta}) = \text{vec}((\boldsymbol{\Theta}\boldsymbol{P}_t) \circ \boldsymbol{\Theta})^\top \mathbf{1}.$$

Putting both penalties together, we obtain the total penalty for the array product as $\text{pen}_s + \text{pen}_t$.

# B    Additional results

## B.1    Simulation study

Figure 6 shows the estimation error of the weight surface using different established methods and our proposed implementation (Neural Network).

All methods perform equally well with minor differences in the weight surface estimation. As expected, neural networks perform better in small signal-to-noise (SNR) settings due to their implicit regularization when trained with stochastic gradient descent and using mini-batches, whereas for large SNR, the full batch routines implemented by boosting and additive models perform slightly better. As becomes apparent in Figure 4(a), these differences are negligible in practice.

## B.2    Application

Figure 7 gives an excerpt of the estimated weight surfaces that provide insights into how the relationship between input features (rows; here acceleration measures from the x-dimension) and outcome joints is estimated.

For example, the trunk acceleration (last row in Figure 7) having a short negative and almost instantaneous effect (dark circle) on the ankle and hip adduction moment (first and second column) is a reasonable result from a biomechanical perspective.

We can further plot the predicted curves by all employed methods (Figure 8) and compare the results with the ground truth. While for some joints the prediction of the semi-structured and deep-only model is very similar, the performance for the subtalar joint is notably better when combining a structured model and the deep neural network to form a semi-structured model.

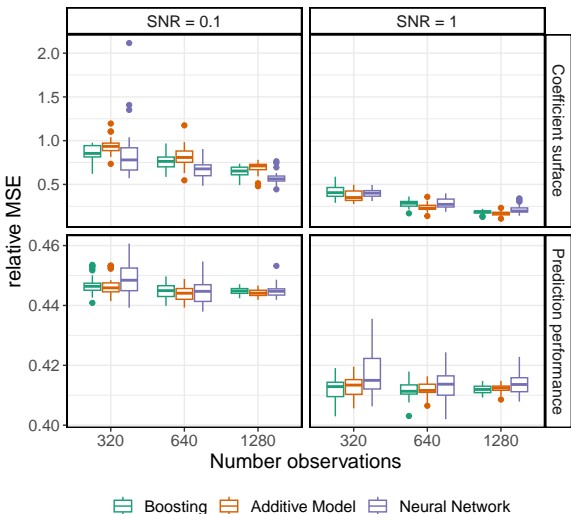

Figure 6: Surface estimation and prediction performance (top and bottom row, respectively) using different optimization methods (colors) for different $n$ (x-axis) and signal-to-noise ratios (SNR; columns).

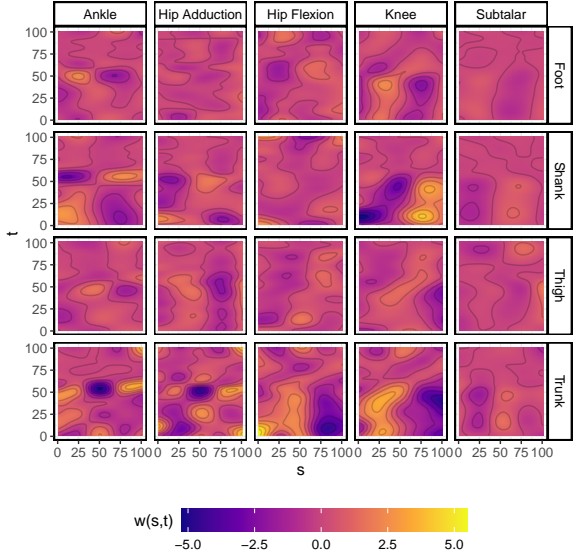

Figure 7: Estimated weight surfaces for different accelerometer predictors (rows) for the different joints (columns). The color of each surface corresponds to the weight a predictor sensor signal at time points $s$ (x-axis) is estimated to have on the $t$th time point (y-axis) of the outcome.

**Comparison with [32]** As requested by an anonymous reviewer, we have also compared our method against the approach suggested in [32]. Compared to [32], we use a different pre-processing of functional predictors which is more in line with our main results. We then run their best model once using a deep-only variant and once using a semi-structured model. The RelRMSE results are given in Table 3, showing that the semi-structured extension works similarly well or in most cases better.

## C Additional experiments

In order to demonstrate our method's applicability beyond biomechanical applications, we run several additional experiments. Datasets come from different fields of application and have different sizes

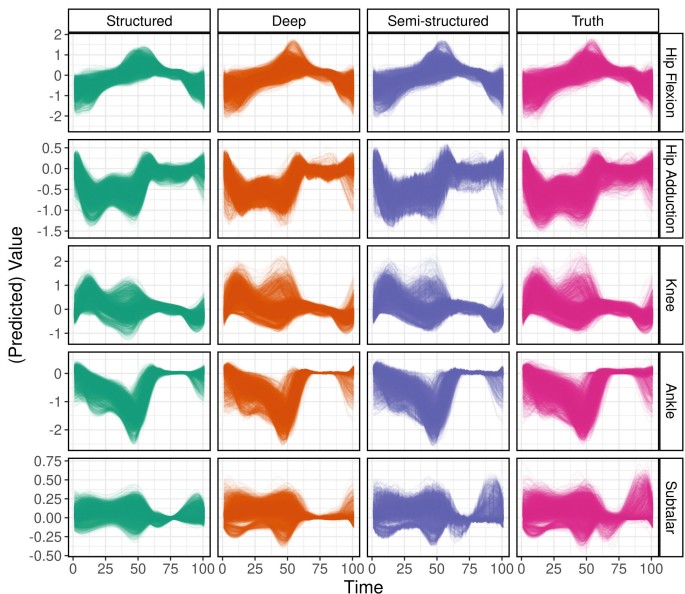

Figure 8: Comparison of predictions of the structured, deep, and semi-structured model as well as the true sensor data (columns) for different joints (rows).

Table 3: Comparison of a plain deep and a semi-structured approach, where the former is based on [32] and the latter its extension to a SSN.

| Measurement | Deep | Semi-str. |
|---|---|---|
| ankle (dim 1) | 0.261 | 0.212 |
| ankle (dim 2) | 0.247 | 0.208 |
| ankle (dim 3) | 0.423 | 0.359 |
| com (dim 1) | 0.054 | 0.048 |
| com (dim 2) | 0.275 | 0.275 |
| com (dim 3) | 0.077 | 0.078 |
| hip (dim 1) | 0.342 | 0.314 |
| hip (dim 2) | 0.301 | 0.300 |
| hip (dim 3) | 0.376 | 0.303 |
| knee (dim 1) | 0.281 | 0.225 |
| knee (dim 2) | 0.318 | 0.270 |
| knee (dim 3) | 0.405 | 0.383 |

(number of observations, number of functional predictors, number of observed time points). We also use these examples to 1) compare the two different architectures suggested in Fig. 2(a) and Fig. 2(b), 2) compare against a non-linear FFR implementation from [40].

For all the additional experiments, the deep network version from Fig. 2(a) uses the same encoding and decoding basis for both $\lambda^+$ and $\lambda^-$ (using $K_j = 20$ basis functions for the $s$-direction and $U = 20$ basis function for the $t$-direction). The trunk of the deep network part $\lambda^-$ is defined as a fully connected neural network with 100 neurons each, followed by a dropout layer with dropout rate of 0.2 and a batch normalization layer. The deep network version from Fig. 2(b) uses the same architecture, but instead of using a shared encoding and decoding part, it simply concatenates the inputs and processes the three-dimensional object until the last layer, which is then reshaped to have the correct dimensions. We have not tuned this architecture, the number of layers, neurons, or dropout rate, as the purpose of these additional experiments is solely to demonstrate that the (SS)FNN approach yields competitive results with only the structured part while the deep part can further help to improve predictive performance.

### C.1 Predicting EMG from EEG signals in cognitive affective neuroscience

The dataset analyzed in [50] contains 192 observations of three different EMG signals from facial muscles used as different outcomes (similar to the different joint moments) and up to 64 EEG signals measured at different parts of the brain. Both EMG and EEG signals are measured at 384 equidistant time points. Following [50], we choose the EXG4-EXG5 signal as output signal and the Fz-, FCz-, POz-, and Pz-electrode as input signals. Based on 10 train/test-splits, we obtain the following table:

Table 4: Model performance comparison

| Model | Rel. RMSE ($\downarrow$) | Correlation ($\uparrow$) |
|---|---|---|
| Boosting | 0.195 (0.137) | 0.184 (0.130) |
| FNN | **0.190** (0.132) | 0.171 (0.111) |
| SSFNN (a) | 0.210 (0.152) | **0.201** (0.162) |
| SSFNN (b) | 0.192 (0.177) | 0.184 (0.287) |

Results confirm again that our FNN implementation leads to similar values as Boosting, while including a deep neural network part can improve the performance.

### C.2 Predicting air quality from temperature curves

The dataset analyzed in [40] contains 355 observations of daily NO2 measurement curves (i.e., the functional domain is the hours 1 to 24) together with four other functional pollutants as well as the temperature and relative humidity. The goal is to predict the NO2 concentration using other pollutants, the temperature, and the humidity. The proposed approach in [40] uses a non-linear function-on-function regression (FFR) which we compare against. The results are given in the following table:

Table 5: Comparison of Model Metrics

| Model | Rel. RMSE ($\downarrow$) | Correlation ($\uparrow$) |
|---|---|---|
| Non-linear FFR | 1.000 (0.000) | 0.878 (0.024) |
| FNN | 1.000 (0.000) | **0.908** (0.009) |
| SSFNN (a) | **0.988** (0.024) | 0.880 (0.052) |
| SSFNN (b) | 0.991 (0.042) | 0.897 (0.047) |

Results suggest that the given task is rather easy and a linear FFR model is already sufficient to obtain good prediction performance. Hence FNN outperforms both the non-linear FFR model and the two semi-structured approaches

### C.3 Predicting hot water consumption profiles in urban regions

The dataset taken from [39] contains hourly hot water consumption profiles (i.e., the functional domain is again 1 to 24) which are available for different administrative districts. The goal of the analysis is to predict the hot water consumption of a specific region given multiple (here 6) far-distant administrative districts. The results are given in the following table:

Table 6: Model Performance Evaluation

| Model | Rel. RMSE ($\downarrow$) | Correlation ($\uparrow$) |
|---|---|---|
| Additive Model | 0.814 (0.006) | 0.761 (0.006) |
| FNN | **0.810** (0.008) | 0.754 (0.010) |
| SSFNN (a) | 0.887 (0.052) | 0.799 (0.042) |
| SSFNN (b) | 0.893 (0.077) | **0.803** (0.037) |

Similar to the first dataset, the FNN achieves similar performance as the reference model while including a deep neural network improves the performance.

# D  Experimental details and computational environment

## D.1  Experimental details

In all experiments, we use the Adam optimizer with default hyperparameters. No additional learning rate schedule was used. The batch size, maximum number of epochs and early stopping patience was adjusted depending on the size of the dataset. A prototypical implementation is available as an add-on package of `deepregression` [51] at `https://github.com/neural-structured-additive-learning/funnel`.

## D.2  Computational environment

All computations were performed on a user PC with Intel(R) Core(TM) i7-8665U CPU @ 1.90GHz, 8 cores, 16 GB RAM using Python 3.8, R 4.2.1, and TensorFlow 2.10.0. Run times of each experiment do not exceed 48 hours.

