# OpenReview forum: "A Functional Extension of Semi-Structured Networks"
_NeurIPS.cc/2024/Conference — NeurIPS 2024 poster_

### Official Review · Reviewer_Ta2r · 2024-07-11

**Soundness:** 3
**Presentation:** 4
**Contribution:** 3
**Rating:** 6
**Confidence:** 3

**Summary:**

This paper extends the application of semi-structured networks to functional data. The orthogonalization technique makes this method more scalable than existing methods.

**Strengths:**

This paper is well-structured and well-written. The idea is easy to follow despite the experiment being conducted in an abstract functional space. The authors have conducted extensive experiments to show the new method's applicability and generalizability.

**Weaknesses:**

The authors emphasize the interpretability of the proposed method, but they have not demonstrated this property in their experiments. Besides, the FFR $\lambda^+$ can only explain the linear part of the target function $y$, but most of the time, we hope to explain the nonlinear interaction in the neural network.

**Questions:**

Q1: The authors claim that FFR can serve as "a functional version of a residual connection "(lines 141-142) and show that the FFR component can empower FNN for better model performance, as shown in Figure 5. Why do we not add the residual design to the FNN directly? Comparing the model performance between FNN, FNN with residual network design, and FNN+FFR in Figure 5 could answer this question.

Q2: Under the discretization setting, what are the differences between time series modeling and functional networks?

**Limitations:**

The limitations are well discussed, but the authors have not stated the GPU specification in Appendix D.2

---

> ### Author Rebuttal · Authors · 2024-08-05
>
> We thank the reviewer for the thoughtful and detailed comments. Below we address the mentioned weaknesses and questions.
>
> -----
>
> ### Weaknesses
>
> > [...] interpretability of the proposed [...] not demonstrated in their experiments
>
> We agree that this is an important part of our work. We would like to point out that we have included result interpretations in Figures 1 and 3, and presented the estimated weight surfaces in Figure 7 in the Appendix, accompanied by a brief explanation. While placing Figure 7 with interpretability results in the Appendix might seem less ideal, it is essential to note that interpreting the estimated weight surfaces of function-on-function regression models is inherently complex. This complexity is further compounded by the need for domain knowledge in biomechanics to fully grasp the results. Therefore, we chose to provide Figure 1 with a toy example and place Figure 7 in the Appendix. Nonetheless, we can confirm that the results are highly plausible from a biomechanical perspective and offer valuable insights into the relationships between accelerations and moments in the analyzed joints.
>
> > Besides, the FFR $\lambda^+$  can only explain the linear part of the target function $y$, but most of the time, we hope to explain the nonlinear interaction in the neural network.
>
> We agree with the reviewer that nonlinearities are not of lesser importance. However, we would like to point out that, although the $\lambda^+$ part of the model is linear in the coefficients of the basis functions, this implies non-linearity in the estimated function-on-function effects. Figures 1, 3, and 7 clearly illustrate this non-linearity, where certain combinations of input and output signals change nonlinearly over the two different time domains. In addition, while it is possible to include interactions between input signals in $\lambda^+$, this increases the complexity of interpretation. We believe it is up to the modeler to decide the appropriate order of interactions to assess in the structured part and what effects to move into the black-box mechanics of the deep network.
>
> -----
>
> ### Questions
>
> > Q1: [...] Why do we not add the residual design to the FNN directly?
>
> We thank the reviewer for this interesting question. While we agree that adding a residual connection directly into the FNN would also improve performance compared to the deep-only network, stacking multiple residual connections would complicate the interpretation. In such a case, we would need to understand the total effect of the structured (linear) part on the outcome in this residual network and, even more challenging, find a way to orthogonalize such a model. Our current proposal addresses this issue clearly, as there is a direct connection from the input to the output that is linear in the coefficients.
>
> > Q2: Under the discretization setting, what are the differences between time series modeling and functional networks?
>
> This is an excellent question that often arises in functional data analysis (FDA). The primary difference between time series modeling (TSM) and FDA is that, in FDA, the curves or “time series” are observed repeatedly and are viewed as replications of the same underlying process (e.g., acceleration curves across the time of one stride). In contrast, TSM typically deals with time series that do not have replications (we cannot observe today’s stock prices multiple times with different noise processes; we only observe each stock price time series once).
> In FDA, we further predict entire curves on the same domain as the training data, while TSM often focuses on forecasting specific future values. Additionally, FDA usually assumes the smoothness of the process (acceleration profiles, at least theoretically, do not have discontinuities) and is concerned with the shape of the curve (e.g., the curvature and course of the signal over time). In TSM, the shapes of the time series are often not the primary focus.
>
>  > [...] the authors have not stated the GPU specification in Appendix D.2
>
> All models were trained on a CPU; we did not use any GPUs.
>
> -----
>
> Please let us know whether we have addressed all weaknesses as well as questions.

---

> > ### Comment · Reviewer_Ta2r · 2024-08-12
> >
> > Thank you for the authors' reply, which addressed much of my concern. This paper and the authors' replies both demonstrated a solid technical and theoretical foundation. However, as the author has mentioned, model interpretability requires domain knowledge in biomechanics to perceive, which refrained the general audience from intuitively understanding the power of the model. Thus, I will maintain my score unchanged.

---

> ### Author Response · Authors · 2024-08-12
>
> Dear Reviewer Ta2r,
>
> Thank you for reading and acknowledging our rebuttal. We would like to briefly comment on the statement
>
> > model interpretability requires domain knowledge in biomechanics to perceive, which refrained the general audience from intuitively understanding the power of the model.
>
> While it is true that the interpretability **of the results** requires domain knowledge (which applies to every field, not just biomechanics), the interpretability of the model itself does not. The coefficient surfaces estimated by our model can be interpreted like any other bivariate effect (similar to those in spatial applications --- however, instead of latitude and longitude, we provide a relationship between $y(t)$ and $x(s)$).
>
> We assume that this is what the reviewer meant, but we wanted to clarify this point to ensure nothing was left unaddressed while we still have the opportunity to respond to comments.
>
> Thank you again for your time reading our paper and providing the review. Please let us know if there are any remaining concerns that we should address.

---

### Official Review · Reviewer_9i2N · 2024-07-11

**Soundness:** 3
**Presentation:** 3
**Contribution:** 2
**Rating:** 6
**Confidence:** 3

**Summary:**

In this paper, the authors develop a semi-structured model for functional data, summing an interpretable linear model with a more general nonlinear functional neural network. The authors validate the improved performance of the combination relative to the individual components on a variety of biomechanics datasets. The authors also discuss how to preserve the interpretability of the semi-structured model with a linear weight surface by projecting the neural network predictions into the subspace of the linear model.

**Strengths:**

The approach provides a natural adaptation of semi-structured models to functional data and is worthy of detailed investigation. Experiments performed on synthetic and real data are reasonably thorough and help validate the claims on efficacy and interpretability. The ability to retain the linear model interpretability using the projection is interesting and specific to the setting.

**Weaknesses:**

The paper accomplishes what it sets out to do fairly well, though the contributions of the paper through providing evidence for new phenomena, novel insights, or new algorithmic ideas seem much more limited.

In terms of the structured part of the model, the linear regression with the functional encoder and decoder bases for function -> function regression, I assume that this approach has been used before but I did not see it in any of the explicit references. Can the authors comment on whether this component is original to their work or whether this component is building on what already exists in the literature?

**Questions:**

One aspect I did not see mentioned in the paper was the actual choice of basis functions for encoder and decoder. These should certainly be specified along with any justification or tuning of hyperparameters involved (e.g. bandwidth parameters?).

Fig2a and 2b present two different potential options for constructing this semi-structured model, might it be worth evaluating the performance of structure a)? Would we expect the performance to be similar, what are the drawbacks?

When fitting the parameters of the linear model, it is mentioned the large computational cost. I did not find the explanation of how the scalable implementation circumvents this to be particularly clear. Also, even for solving the linear system could one not use more sophisticated matrix free methods such conjugate gradients with the kronecker product matrix, or for example SVRG which would converge quickly even when the number of data points and features is large?

**Limitations:**

I think the limitations are sufficiently discussed.

---

> ### Author Rebuttal · Authors · 2024-08-05
>
> We thank the reviewer for the thoughtful comments and pointing out ways to improve our manuscript. Below we address the mentioned weaknesses and questions.
>
> -----
>
> ### Weaknesses
>
> > Can the authors comment on whether this component is original to their work [...]
>
> We thank the reviewer for bringing this up. While a functional autoencoder has been proposed in the past (e.g., Hsieh et al., 2021) and most recently by Wu et al. (2024), these studies focus on the representation learning (and reconstruction) of the same signal. Other papers focusing on function-on-function regression (e.g., Luo and Qi, 2024; Rao and Reimherr, 2023; Wang et al., 2020) suggest similar approaches to ours but without the option to jointly train a structured model and a deep network.
>
> We also believe that our idea to encode and decode signals using a general basis representation is to some extent novel. However, any mapping from function to scalar values and back (such as the functional PCA in Wang et al., 2020) can potentially be interpreted as such an encoding strategy. We will clarify this distinction and our contribution, and we thank the reviewer for highlighting this point.
>
> -----
>
> ### Questions
>
> > [...] choice of basis functions for encoder and decoder. These should certainly be specified [...]
>
> We thank the reviewer for pointing this out. In our experiments, we used thin plate regression splines but also obtained similar results with B-splines of order three and first-order difference penalties. We will provide this information in an updated version and will add a short discussion of possible other options with their pros and cons.
>
> > Fig2a and 2b present two different potential options for constructing this semi-structured model, might it be worth evaluating the performance of structure a)? Would we expect the performance to be similar, what are the drawbacks?
>
> This is indeed an important question. We have included such a comparison in Appendix C in the original version of the paper, where we “compare the two different architectures suggested in Fig. 2(a) and Fig. 2(b)”. Using different real-world datasets, we found that both architectures perform quite similarly. In practice, our biomechanics project partners might be in favor of option b) as it allows them to use an established network for the deep model part and gives more flexibility, whereas from a practical point of view, option a) will be potentially more parameter-sparse and allows to jointly learn the function embedding, which in turn increases interpretability.
>
> > When fitting the parameters of the linear model, it is mentioned the large computational cost. I did not find the explanation of how the scalable implementation circumvents this to be particularly clear.
>
> We thank the reviewer for this question. In Section 3.4, we have provided an explanation of how our implementation solves this problem with a focus on the space complexity (memory costs). In particular,
>
> - memory costs are reduced by using mini-batch training and not having to cast our model into long-format as done by classical approaches. Our approach could indeed then be combined with the SVRG approach by Johnson and Zhang as the reviewer suggested.
> - In contrast to classical implementations, we rely on array computations (as e.g. given in Equation 8 in the paper). Using the array model formulation, we never have to explicitly construct the Kronecker product of the basis matrices, which saves additional memory.
> - In addition to the two previous solutions, we recycle the basis functions in the s-direction, which is non-trivial for other software implementations. In the neural network, however, this “simply” boils down to having multiple connections to the same matrix object in the computational graph.
>
>
> -----
>
> ### References
>
> Hsieh, T. Y., Sun, Y., Wang, S., & Honavar, V. (2021). Functional autoencoders for functional data representation learning. In Proceedings of the 2021 SIAM International Conference on Data Mining (SDM) (pp. 666-674). Society for Industrial and Applied Mathematics.
>
> Luo, R., & Qi, X. (2024). General Nonlinear Function-on-Function Regression via Functional Universal Approximation. Journal of Computational and Graphical Statistics, 33(2), 578-587.
>
> Rao, A.R. and Reimherr, M., 2023. Modern non-linear function-on-function regression. Statistics and Computing, 33(6), p.130.
>
> Wang, Q., Wang, H., Gupta, C., Rao, A.R. and Khorasgani, H., 2020, December. A non-linear function-on-function model for regression with time series data. In 2020 IEEE International Conference on Big Data (Big Data) (pp. 232-239). IEEE.
>
> Wu, S., Beaulac, C. and Cao, J., 2024. Functional Autoencoder for Smoothing and Representation Learning. arXiv preprint arXiv:2401.09499.

---

### Official Review · Reviewer_SfMu · 2024-07-12

**Soundness:** 3
**Presentation:** 3
**Contribution:** 1
**Rating:** 6
**Confidence:** 2

**Summary:**

An extension to semi structured networks is introduced that combines a traditional linear and interpretable model component $\lambda^+$ with a flexible $\lambda^-$ neural network component to better approximate relations in biomechanical application. The model performs equally to traditional methods but requires less memory and remains interpretable thanks to a post-hoc orthoganalization.

**Strengths:**

_Originality:_ The introduced method nicely combines established approaches and compares it to some, but not to all relevant related articles (see details under weaknesses).

_Quality:_ The submission is technically sound in that most claims are well supported by experimental results; some exceptions are listed under weaknesses. The paper is nicely structured but, for my taste, lacks an intuitive examples and explanations to nudge and inform readers that are unfamiliar with the topic. An exhaustive number of experiments is conducted on several problems, both on synthetic and real-world data, allowing a quantitative assessment of the introduced method.

_Clarity:_ Although authors could add more intuition about the effect and function of certain (mathematical) concepts, the manuscript is well organized and clearly written. In particular, the authors very nicely outline their hypothesis and refer to them in the results section. Apart from some aspects detailed in the weaknesses, the manuscript informs the reader adequately. The submission is complemented with code, yet the data to reproduce experiments is lacking (due to space constraints).

_Further comments_:
- Figure 2 nicely illustrates the two proposed implementation options for hybrid models.
- Figure 3 is helpful, but I find it hard to understand entirely. What exactly does the x-axis represent and are the values normalized to [0, 100] or what is the data range and what does it represent? The last sentence in the caption is not clear to me, i.e., when stating "... early time points $s$". This is probably formulated suboptimally, as $s$ is reported to relate to a sensor signal and $t$ to time.

**Weaknesses:**

_Significance_: Even though the authors emphasize their method's superiority in lines 144-146, the performance improvement over existing approaches is rather mediocre and does not seem significant, in particular when considering the comparably large standard deviations. The error distributions of the different methods seems to overlap to large extend. In this vein, it would be appreciated if error bars were reported in Figure 5. In that figure, you may consider to flip the y-axis to make it more intuitive and report the relative error reduction over the baseline.

_Further comments_:
- It took me very long to get a grasp of the concept of functional data/analysis/regression. In fact, until the very end of the manuscript, I did not fully understand the actual task at hand and permanently felt somewhat lost. Authors may give a quick practical example of functional analysis early to catch readers sooner. Even the example in lines 109--123 did not resolve my uncertainty about where such processes find application. I thus suggest to add one clear and illustrative practical example and spend more effort to introduce the problem at the beginning of the introduction rather than starting with related work.
- While novelty for functional data analysis seems reasonable, I cannot find a substantial contribution to the machine and deep learning community (given the paper is submitted to a top tier ML conference). Even though I like the paper's argumentation and substantiation and perceive it as a generally sound work, the manuscript might better be suited to a journal that deals with functional analysis or with biomechanics.
- Claim in l265 not supported with experimental results. Please provide error and dispersion metrics (such as in Table 1 and 2) to allow a quantitative comparison of the methods.
- Typo in line 331: Either "reveal that" or "yields"?

**Questions:**

1. What are the additive structures you are referring to in l15, can you provide some examples?
2. What is the proportional contribution of $\lambda^+$ and $\lambda^-$ to the overall solution? How is guaranteed that $\lambda^-$ is not solving the entire problem solely, thus loosing all interpretability? The orthogonal formulation seems to take a step into that direction, but I am not sure what the orthogonal formulation actually does, and if the patterns revealed after its application are helpful and realistic.
3. What are the integration weights $\Xi$ concretely in line 182 and what are they required for (can they be dropped for simplicity yielding a conventional loss function)?
4. The orthogonalization approach in Section 3.3 appears very beneficial, yet, frankly, I could not follow it as the math is quite abstract. Can you share an intuition what the orthogonalization implements and how it effects $\lambda^-$ to be less dominant, if so? Is it some kind of regularization to minimize its contribution (or how would, e.g., L2 regularization compare)?
5. Relating to Figure 4, how does runtime compare for the three implementations for $\lambda^-$? Given the memory is capped for the batched neural network, it should take multiple evaluations to process the same amount of data. How does the memory compare if the other methods are implemented in a batched fashion too?
6. Can you share an intuition of why the neural network performs worse in high SNR regimes? I'd expect a neural network be superior in all conditions, given it is sufficiently large and regularized appropriately.
7. Could you include the results from reference [24] into your results in Section 4.2.1 to allow a comparison between your and an established method? It is crucial to assess how your approach performs in comparison to existing methods.

**Limitations:**

Limitations are addressed by the authors.

---

> ### Author Rebuttal · Authors · 2024-08-05
>
> We thank the reviewer for the detailed and thorough review of our manuscript.
>
> -----
>
> ### Weaknesses
>
> > improvement [...] rather mediocre [...] would be appreciated if error bars were reported
>
> We only report results for a single train/test split as the application fixes this. To still provide a measure of uncertainty, we show individual joint performances in Fig. 5, and Tab. 2 summarizes findings with standard dev., showing a significant improvement for semi-structured models. However, it's not only about performance improvement but also being able to quantify the explained proportions by the structured and deep part.
>
> > I cannot find a substantial contribution to the [ML/DL] community [...] better be suited to a [biomechanics] journal
>
> We thank the reviewer for the comment but politely disagree. 1) The paper is too technical for a biomechanics journal (a manuscript focusing on interpretation will be submitted to such a journal). 2) We believe that projects with a motivating application should be valued at least as much as those proposing methods without real applications. 3) We also think that we provide a substantial contribution to the ML/DL community as functional data is well represented at NeurIPS (e.g., Boschi et al., 2021; Gonschorek et al., 2021; Madrid Padilla et al., 2022), ICML (e.g. Yao et al., 2021; Heinrichs et al., 2023), etc., and our paper advances both orthogonalization and semi-structured models, also discussed in these communities (e.g., Lu et al., 2021; Vento et al., 2022; Rügamer et al., 2023, 2024).
>
> > Claim in l265 not supported with experimental results. Please provide error and dispersion metrics
>
> We provide these metrics as boxplots in Fig. 6 (Appendix).
>
> ### Questions
>
> > What are the additive structures [...] provide some examples
>
> We wrote: “For example, neural additive models [...] such as generalized additive models”. We will revise the text to make these examples more clear.
>
> > proportional contribution of  $\lambda^+$  and $\lambda^-$ [...] How [..] $\lambda^-$ is not solving the entire problem
>
> The proportional contribution will depend on the application. $\lambda^-$ is not solving the entire problem due to the orthogonalization (see answer below).
>
> > What are the integration weights $\Xi$ concretely [...] can they be dropped
>
> The integration weights are used to approximate the integral (trapezoidal Riemann weights in our paper). For time points on an equidistant grid, one could potentially drop them.
>
> > intuition what the orthogonalization implements [...] Is it some kind of regularization
>
> The orthogonalization process subtracts everything that could have been explained through $\lambda^+$ from $\lambda^-$ and adds it to $\lambda^+$, ensuring that the two parts are orthogonal. This concept can be understood by analogy to applying multiple regression models. First, generate predictions by regressing the actual outcome on the features via $\lambda^+$ and $\lambda^-$. Next, regress these predictions on $\lambda^+$ to determine the portion that can be explained by $\lambda^+$. The residual term, representing what cannot be explained by the structured part, will be orthogonal to this part. The orthogonalization employs this approach using appropriate projection matrices and can be applied whenever the structured part is linear in the coefficients. While it may not be straightforward to see how this works for our model, Section 3.3 demonstrates how to do this using vector operations. It is not a regularization in the classical sense, as it exactly enforces orthogonality between the two parts.
>
> > Relating to Figure 4, how does runtime compare for the three implementations for $\lambda^-$?
>
> Figure 4 investigates whether “without the **deep part**, [...] can [we] recover complex simulated function-on-function relationships [...] while scaling better than existing approaches”. Hence, no deep part was included in this experiment. Also, neither the boosting nor the additive model approach would allow the incorporation of the $\lambda^-$ part.
>
> > How does the memory compare if the other methods are implemented in a batched fashion
>
> This would result in improved memory consumption w.r.t. number of observations, however, not w.r.t. the number of functional predictors.
>
> > intuition of why the neural network performs worse in high SNR regimes?
>
> Full-batch optimization (additive model, boosting) is beneficial when there is less noise in the data (and vice versa, the neural network's stochastic optimization induces regularization and might thereby better deal with noise).
>
> > Could you include the results from reference [24]
>
> As [24] used a different processing of functional predictors, we first had to adapt their code. We then ran their best model once using a deep-only variant and once using a semi-structured model. RelRMSE results are as follows:
>
> | | Deep | Semi-str. |
> |-|-|-|
> | ankle (dim 1) | 0.261 | 0.212 |
> | ankle (dim 2) | 0.247 | 0.208 |
> | ankle (dim 3) | 0.423 | 0.359 |
> | com (dim 1)| 0.054 | 0.048 |
> | com (dim 2) | 0.275 | 0.275 |
> | com  (dim 3) | 0.077 | 0.078 |
> | hip (dim 1) | 0.342 | 0.314 |
> | hip (dim 2) | 0.301 | 0.300 |
> | hip (dim 3) | 0.376 | 0.303 |
> | knee (dim 1) | 0.281 | 0.225 |
> | knee (dim 2) | 0.318 | 0.270 |
> | knee (dim 3) | 0.405 | 0.383 |
>
> -----
>
> ### References
>
> - Boschi et al., 2021. A highly-efficient group [...] NeurIPS.
> - Gonschorek et al., 2021. Removing inter-experimental [...] NeurIPS.
> - Heinrichs et al., 2023, Functional Neural Networks [...] ICML.
> - Lu et al., 2021. Metadata normalization. CVPR.
> - Madrid Padilla et al., 2022. Change-point detection [...] NeurIPS.
> - Rügamer, 2023. A new PHO-rmula [...] ICML.
> - Rügamer, et al., 2024. Generalizing Orthogonalization [...] ICML.
> - Vento et al., 2022. A penalty approach [...] MICCAI.
> - Yao et al., 2021. Deep learning for functional [...] ICML.
>
> -----
>
> Please let us know whether we have addressed all points and whether these answers are sufficient to re-evaluate your score.

---

> > ### Comment · Reviewer_SfMu · 2024-08-09
> > **Major concerns resolved**
> >
> > Thanks for your efforts in providing additional results, in particular verifying study [24] in your experimental setup, and for justifying your study's suitability to NeurIPS. I must admit that I am not an expert of this field and still have difficulties in assessing the relevance, but your arguments appear sound to me.
> >
> > The additional results provided in your rebuttal are convincing. They help me to further understand where your model is superior; A thorough interpretation and discussion of these results would be helpful for readers. That is, for what reason does you semi-structured approach boost performance in some cases but not in others?
> >
> > Assuming the explanations and results from the rebuttal are carefully woven into the manuscript, I am raising my score from 4 to 6 but lower my confidence further from 3 to 2, as I do not feel qualified to provide a strong and well justified assessment of this work. Good luck!

---

> > > ### Author Response · Authors · 2024-08-09
> > >
> > > Dear Reviewer SfMu,
> > >
> > > Thank you for your thoughtful feedback and for taking the time to review our rebuttal. We appreciate that you adjusted the score in response to our rebuttal. We will ensure that all explanations and results from the rebuttal are included in the revised version of our manuscript. Additionally, we will dedicate a separate section to discussing the reasons why the semi-structured approach boosts performance in some cases but not in others. As stated in the general rebuttal, we believe your review has allowed us to further improve our paper through additional clarifications and experiments, and we are thankful for this input.

---

### Official Review · Reviewer_DyRG · 2024-07-15

**Soundness:** 3
**Presentation:** 2
**Contribution:** 2
**Rating:** 5
**Confidence:** 2

**Summary:**

The paper proposes a hybrid approach that combines the benefits of neural networks with those of more structured models (generalised additive models). They show benefits on real and simulated data.

**Strengths:**

- The proposed idea makes sense and is novel AFAIK
- The empirical results show good performance compared to either purely deep models or purely structured models
- The description of the method in sections 3.1 to 3.4 is clear.

**Weaknesses:**

- How much tuning has been done for the pure deep-only network baseline from Section 4.2.2? I would be a bit worried that the findings are very dependent on the choice of architecture of this model? E.g. does it have any increased capacity over the deep part of the semi-structured network? And does this architecture have suitable inductive biases for fitting linear functions (maybe residual connections which add a linear transformation of the input to the output)?
- The presentation could be made clearer in a couple of areas. In particular, (1) Equation 3 is the first time that the “double function call” notation of e.g. $\lambda^-(X)(t)$ is used. Having a double function call makes sense given that there is a function which takes X as input and outputs another function. However it is not clear to me why this is then not used in Equation 1, where you could presumably replace $\mu(t)$ with $\mu(X)(t)$? And e.g. in the last part of Section 2.2, why write $h^{(L)}(t)$ instead of $h^{(L)}(X)(t)$? Would it be possible to make this clearer? And also (2), when functional data analysis is introduced in Section 1 and 2, it would be helpful if an example application was described

**Questions:**

See weaknesses. I am willing to increase my score if these are addressed.

**Limitations:**

Adequately addressed

---

> ### Author Rebuttal · Authors · 2024-08-05
>
> The reviewer raises two points that we answer below. We thank the reviewer for these thoughtful comments, but in both cases would like to point out that these "weaknesses" have already been addressed in our original submission.
>
> -----
>
> > How much tuning has been done for the pure deep-network baseline? I would be a bit worried that the findings are very dependent on the choice of architecture of this model?
>
> We agree that this is an important aspect that needs to be taken into account. Fortunately, some already-tuned architectures are widely adopted in biomechanics. As we write in the beginning of Section 4 “we use a tuned InceptionTime [16] architecture from the biomechanics literature [ 24]. As FFR and boosting provide an automatic mechanism for smoothing, no additional tuning is required.”. In other words, we don’t tune any of the models. While tuning might certainly benefit one or the other model, we think that it is difficult to make a perfectly fair comparison between all methods. In this light, we adopt a strategy that is closest to what practitioners in the field would likely do — use the tuned Inceptionnet with the hyperparameters found to be optimal in previous studies and take this architecture without further modifications.
>
> We will make this more clear in a revised version of the manuscript.
>
> > The presentation could be made clearer. E.g. when $\lambda^+$  and $\lambda^-$  are introduced, they should be more clearly defined.
>
> We thank the reviewer for this comment. We have defined $\lambda^+$ in the paragraph right after it was introduced ($\lambda^+(t) = \sum_{j=0}^J \lambda_j^+(t)$ with $\lambda_j^+(t) = \int_{\mathcal{S}_j} w_j(s,t) x_j(s)ds)$ and in the following paragraph (“Deep model part”) discuss choices for $\lambda^-(t)$.
> We are happy to include further details if there is anything else the reviewer is missing.
>
> -----
>
> Please let us know whether we have addressed all weaknesses and whether these answers are sufficient to re-evaluate your score.

---

> ### Comment · Reviewer_DyRG · 2024-08-07
> **Updated review**
>
> Hi, please see my updated review which hopefully makes it clearer what needs to be done for me to recommend acceptance

---

> ### Author Response · Authors · 2024-08-07
>
> Dear Reviewer DyRG,
>
> Thank you very much for reading our rebuttal and your prompt response. We also appreciate the revised review, which will help us to improve the clarity of our paper. Below we respond to the raised questions point-by-point:
>
> > How much tuning has been done for the pure deep-only network baseline from Section 4.2.2?
>
> That is a very good question, and we can understand your concerns well. However, note that our previous response also applies to the application in Section 4.2. The quoted sentence “we use a tuned InceptionTime [16] architecture from the biomechanics literature [24]” is given in Section 4 and applies to both subsections. In other words, we did not tune the deep network part (for both the deep-only and the semi-structured model). We will make this more clear in a revised version of the manuscript.
>
> > I would be a bit worried that the findings are very dependent on the choice of architecture of this model? E.g. does it have any increased capacity over the deep part of the semi-structured network?
>
> This is another good question that the reviewer raises. Note, however, that the deep network is the same for the “deep-only” model and the semi-structured network.
>
> > And does this architecture have suitable inductive biases for fitting linear functions (maybe residual connections which add a linear transformation of the input to the output)?
>
> Good point. We can confirm that the architecture is suitable for fitting linear functions. A large part of the variation in these applications is explained by the (linear) function-on-function regression, which the InceptionTime model can also capture. This is also apparent from the fact that the orthogonalization has a large impact on the explained relationship (Figure 3), from predictions in Figure 8 in the Appendix and further confirmed in [24], where the deep network can represent a similar function space as the function-on-function regression (at least for this specific application).
>
> We will make these points more clear in a revised version and thank the reviewer again for bringing up these points.
>
> > The presentation could be made clearer in a couple of areas. In particular, (1) Equation 3 is the first time that the “double function call” notation of e.g. $\lambda^-(X)(t)$ is used. Having a double function call makes sense given that there is a function which takes X as input and outputs another function. However it is not clear to me why this is then not used in Equation 1, where you could presumably replace $\mu(t)$ with $\mu(X)(t)$?
>
> We thank the reviewer for pointing this out. The reviewer is absolutely correct and $\mu$ is also a double function with the first argument being the data $X$. In line 101, we defined $\mu$, writing “An FFR for the expected outcome $µ(t) := \mathbb{E}(Y(t)|X=x)$" and explicitly dropped the dependence (by considering $X$ to be fixed with realization $x$). We agree, however, that this might lead to confusion and will add the second argument where appropriate to be consistent.
>
> > And e.g. in the last part of Section 2.2, why write $h^{(L)}(t)$ instead of $h^{(L)}(X)(t)$? Would it be possible to make this clearer?
>
> Again, the reviewer is correct. We simply tried not to overload the presentation but agree that this should be consistent and will update the notation. Thank you for pointing this out.
>
> > And also (2), when functional data analysis is introduced in Section 1 and 2, it would be helpful if an example application was described
>
> We agree and will add an example from the field of biomechanics (predicting ground-reaction forces with acceleration curves) and neuroscience (predicting muscle movements from brain signals).

---

> > ### Author Response · Authors · 2024-08-12
> >
> > Dear Reviewer DyRG,
> >
> > Thank you once again for your thoughtful comments and for clarifying the criteria for recommending acceptance. We hope that our previous response has addressed all your questions and would like to know if there is anything else we can provide at this point.

---

> > ### Comment · Reviewer_DyRG · 2024-08-13
> >
> > Thank you for the rebuttal. Given the updates you say you will make to the paper, I have raised my score to a 5.

---

> ### Author Response · Authors · 2024-08-14
>
> Dear Reviewer DyRG,
>
> Thank you for following up on our response to your revised rebuttal. We appreciate that you have raised the score above the acceptance threshold and clearly outlined what was required from us to earn your recommendation for acceptance. We thank you again for your suggestions to change the notation, add details on experimental details, and include an illustrative example. We will incorporate these into the revised manuscript, which we believe will further strengthen our paper.

---

### Author Rebuttal · Authors · 2024-08-06

We thank reviewers DyRG, SfMu, 9i2N, and Ta2r for their detailed and thoughtful comments. We appreciate your efforts and believe your reviews have allowed us to further improve our paper. We think that have addressed all points raised and eliminated all uncertainties. In detail:

1. **[Reviewer DyRG](https://openreview.net/forum?id=WJAiaslhin&noteId=22uGAaOUUW)**:
    - We clarified uncertainties regarding model tuning and
    - highlighted existing text passages that answer questions about network definitions.
2. **[Reviewer SfMu](https://openreview.net/forum?id=WJAiaslhin&noteId=1jH7ELJVbj)**:
    - We argued that our work is suitable for the ML/DL community,
    - pointed to existing results in the appendix that might have been overlooked,
    - clarified various points regarding additive structures, proportional contributions, integration weights, orthogonalization, and runtime comparisons, and
    - added a comparison study with another approach from the literature as requested.
3. **[Reviewer 9i2N](https://openreview.net/forum?id=WJAiaslhin&noteId=ScrPcIafBW)**:
    - We clarified our contributions,
    - discussed the choice of basis functions, and
    - pointed to additional results in the appendix that might have been overlooked.
4. **[Reviewer Ta2r](https://openreview.net/forum?id=WJAiaslhin&noteId=es1ujpx7su)**:
    - We explained where interpretability is demonstrated in our results and
    - clarified questions about residual connections, non-linearity in the model, and the difference from time series modeling.

-----

Please let us know if there are any further questions or if additional clarifications are needed. We are happy to discuss any open points in the next discussion phase.

---

### Decision · Program_Chairs · 2024-09-25

**Decision:**

Accept (poster)

**Comment:**

The paper proposes a functional extension of semi-structured networks, consisting of an additive combination of an interpretable function-on-function regression and an expressive functional neural network. The paper also addresses questions regarding scalability and identifiability, and demonstrates the effectiveness of the proposed architecture in synthetic scenarios and on real-world biomechanics data.

This paper received reviews on the positive side, with all reviewers recommending acceptance after the rebuttal phase. The reviewers commended the paper’s clear presentation, novel approach, and competitive performance. The orthogonalization approach, in particular, was highlighted as an interesting solution to the identifiability problem in architectures with an interpretable and an expressive component.

Initial concerns expressed by the reviewers included accessibility/relevance and experiment design/results. Multiple reviewers felt that a more thorough introduction to functional data analysis and biomechanics, supported by figures and examples, would improve accessibility/relevance of the manuscript. The authors promised and are expected to update the relevant sections. Initial feedback on the experiment design/results was mixed, with reviewers mentioning marginal improvements over baselines, missing baselines, and limited analysis of interpretable results. The authors responded well to these concerns, providing scalability arguments, additional results, and pointers to interpretable weight surfaces.

The majority of reviewers felt that their concerns were adequately addressed during the rebuttal phase and multiple reviewers increased their rating as a result. Based on the paper’s contributions to the field of semi-structured networks, and with the expectation that the feedback received and clarifications provided during the discussion period will be integrated, the paper is recommended for acceptance as a poster.